# Maneuvering Performance in the Colonial Siphonophore, *Nanomia bijuga*

**DOI:** 10.3390/biomimetics4030062

**Published:** 2019-09-05

**Authors:** Kelly R. Sutherland, Brad J. Gemmell, Sean P. Colin, John H. Costello

**Affiliations:** 1Oregon Institute of Marine Biology, University of Oregon, Eugene, OR 97402, USA; 2Department of Integrative Biology, University of South Florida, Tampa, FL 33620, USA; 3Whitman Center, Marine Biological Laboratory, Woods Hole, MA 02543, USA; 4Marine Biology/Environmental Sciences, Roger Williams University, Bristol, RI 02809, USA; 5Biology Department, Providence College, Providence, RI 02908, USA

**Keywords:** turn, reverse, agility, maneuverability, propulsion, *Nanomia bijuga*

## Abstract

The colonial cnidarian, *Nanomia bijuga*, is highly proficient at moving in three-dimensional space through forward swimming, reverse swimming and turning. We used high speed videography, particle tracking, and particle image velocimetry (PIV) with frame rates up to 6400 s^−1^ to study the kinematics and fluid mechanics of *N. bijuga* during turning and reversing. *N. bijuga* achieved turns with high maneuverability (mean length–specific turning radius, R/L = 0.15 ± 0.10) and agility (mean angular velocity, ω = 104 ± 41 deg. s^−1^). The maximum angular velocity of *N. bijuga*, 215 deg. s^−1^, exceeded that of many vertebrates with more complex body forms and neurocircuitry. Through the combination of rapid nectophore contraction and velum modulation, *N. bijuga* generated high speed, narrow jets (maximum = 1063 ± 176 mm s^−1^; 295 nectophore lengths s^−1^) and thrust vectoring, which enabled high speed reverse swimming (maximum = 134 ± 28 mm s^−1^; 37 nectophore lengths s^−1^) that matched previously reported forward swimming speeds. A 1:1 ratio of forward to reverse swimming speed has not been recorded in other swimming organisms. Taken together, the colonial architecture, simple neurocircuitry, and tightly controlled pulsed jets by *N. bijuga* allow for a diverse repertoire of movements. Considering the further advantages of scalability and redundancy in colonies, *N. bijuga* is a model system for informing underwater propulsion and navigation of complex environments.

## 1. Introduction

Planktonic marine organisms navigate three-dimensional space to acquire food, avoid predation and reproduce. The colonial siphonophore, *Nanomia bijuga*, is a cnidarian with multiple swimming units (nectophores) that can swim forward, in reverse, and turn. The presence of multiple jetting units that can be operated individually or simultaneously opens a wider array of swimming maneuvers than is available to organisms that have only one propulsive unit as in medusan jellyfish. Being colonial allows organisms with simple neurocircuitry and morphology to achieve complex movements and is therefore of direct application to designing multi-jet vehicles that are adept at navigating the ocean.

Pulsed jets have been shown to be effective for generating thrust more efficiently than steady jets [1] and have thus become models for underwater vehicles [2] and soft swimming robots [3]. One of the appealing elements of pulsed jets is they allow for maneuvers in small spaces at low speeds more effectively than propellers [4]. Previous biomimetic designs have emulated single-jetters like jellyfish and squid. The presence of multiple jets along the colony axis in *N. bijuga* can inspire new underwater vehicles that are streamlined (Figure 1a), effective at long-distance cruising, and highly maneuverable due to the strategic placement of jets along the colony axis to produce torque.

*N. bijuga* first gained the attention of oceanographers because it undertakes long vertical migrations on a diel basis and, owing to the gas-filled pneumatophore, is an important component of the sound scattering layer in much of the worlds’ oceans [5]. Detailed studies in the laboratory revealed that in addition to long-distance migrations, *N. bijuga* is also capable of responding to stimuli (light, mechanical disturbances) by executing rapid maneuvers in three-dimensional space, including turning and reversing [6].

Forward swimming and turning in *N. bijuga* rely on coordination of multiple swimming units, which are called nectophores. Because there are multiple nectophores in a linear array, the position and orientation of each unit influences its propulsive role [7]. Larger nectophores at the base of the colony are oriented downward and generate most of the straight-swimming thrust. Newer nectophores that have been recently budded at the apex of the colony have a long lever arm and a high angle relative to the colony axis; these nectophores generate torque for turning and very little thrust. At the level of individual nectophores, the integration of motion by the nectophore and velum—a funnel-shaped band of tissue at the jet orifice—direct the fluid to control jetting and refill and allow for high-speed (~1 m s^−1^), narrow jets (1–2 mm) during forward swimming [8].

Maneuvering in three dimensions is achieved through a combination of placement and coordination of the nectophores and changes in velar orientation. At the nectophore-level, the kinematics of jetting and refill are identical during forward swimming and turning. However, during forward swimming, all or most of the nectophores are recruited to swim either asynchronously (steady-state swimming) or synchronously (escape swimming) whereas during turning, typically only one apical (anterior) nectophore fires, leveraging torque due to its position and orientation [7]. Reverse swimming is achieved by the highly maneuverable velum, which directs the fluid in the anterior direction during reversals ([6]; Figure 1d, Appendix A). As in forward swimming, all or most of the nectophores are recruited during reverse swimming. Because reverse swimming is executed as an escape maneuver, all nectophores fire synchronously.

Previous studies have quantified forward swimming performance in *N. bijuga* [7] but turning and reverse swimming have only been described qualitatively [6]. In the present study, our goal was to: (1) quantify forward and reverse swimming performance and (2) to investigate how individual nectophore kinematics, especially in the velum, control fluid motion and contribute to turning and reverse swimming in *N. bijuga*. Currently, machines cannot match the swimming maneuvers of animals. Quantification of maneuvering performance variables allows for comparisons with other organisms and creates a baseline for improving agility and maneuverability in underwater vehicles. *N. bijuga* is a model for achieving complex maneuvers with simple, distributed morphology and primitive neuronal control. Though studying fast-swimming organisms with narrow (1–2 mm) high speed jets (1 m s^−1^; [8]) is not without challenges, their transparent bodies provide a unique opportunity for fluid imaging.

## 2. Materials and Methods

*N. bijuga* colonies were collected in individual containers from docks at Friday Harbor Laboratories, WA, USA, in June 2014, 2016 and 2017 and maintained in running seawater tables at field temperatures (10–12 °C). Over the course of the study over 100 colonies were collected. We obtained 21 turning or reverse swimming sequences from 16 colonies that were in focus and behaving normally. Colonies used in analyses had a mean nectosome length of 13.9 ± 4.0 mm (mean ± SD) and 9 ± 2.7 nectophores. Nectophores were 3.6 ± 0.46 mm in width. Kinematic and fluid mechanic measurements were made with high-speed videography in custom glass vessels within 24 h of collection [8].

### 2.1. Fluid Mechanics

To examine the relationship between velum kinematics and fluid mechanics of the jet, we used a combination of laser sheet PIV [9] and particle tracking [7]. Images were collected with high speed monochrome video cameras at 500–6400 frames per s (models from Photron Fastcam, Phantom and Edgertronic). For the PIV set-up, 10 *μ*m hollow glass beads were seeded into the tank and illuminated with a <1 mm thick continuous laser light sheet (532 nm). Image sequences during jetting and refill were selected where the velar aperture was bisected by the laser sheet. Image pairs were subsequently analyzed to examine refill using a cross-correlation PIV algorithm with a multi-pass interrogation window size of 64 × 64 pixels down to 32 × 32 pixels and 50% overlap (DaVis 8.3) to produce instantaneous velocity vectors and vorticity contours. Instantaneous jet velocities were too high to capture even at shutter speeds of 1/10,000 s. Therefore, jet velocities were measured by tracking individual particles from image stacks or measuring particle streaks (ImageJ, NIH, Bethesda, MD, USA) [7].

### 2.2. Kinematics

Whole-colony swimming speeds and kinematics were measured from PIV images where the colony remained in-plane over 1–3 pulse cycles. For turning sequences, this method ensured that motion was in two dimensions to extract accurate angular velocities. For kinematic measurements of individual nectophores, *N. bijuga* colonies were illuminated using a brightfield set-up with a 10× LWD objective [10]. Image stacks were imported into ImageJ to measure morphometric variables, swimming speed, time spent jetting and refilling during each pulse cycle and turning parameters (see below).

### 2.3. Turning Performance

Radius of curvature, R—also defined as maneuverability—was calculated by tracking the x, y position of the center of rotation during a turn using ImageJ (Figure 1b; Appendix A). The x, y positions were then fit with a circle of radius R based on a least squares approach using the MATLAB routine Curv.m. Length specific turning radius, R/L, is the radius of curvature divided by the nectosome length.

Instantaneous angular velocity through a turn, ω, also defined as the agility, was calculated by measuring the change in the colony angle, θ, between successive time steps. Colony angle, θ, was defined based on the x, y position of colony tip (just below pneumatophore) at two successive time steps and the center of rotation at the base of the nectosome (Figure 1b). Angles were calculated using a custom MATLAB routine based on the dot product and length of the two vectors that defined the angle. The angular velocity, ω, was calculated as the change in instantaneous angle between time steps. Angular acceleration was calculated as the change in instantaneous velocity between time steps.

## 3. Results

High-speed microvideography with kinematic and fluid mechanical analyses with 21 sequences from 16 *N. bijuga* colonies showed how turns and reversals were executed.

### 3.1. Turning

Turns in *N. bijuga* colonies were accomplished by pulsing of a single apical nectophore. Nectophore kinematics and resultant fluid mechanics during turning matched those of straight swimming and are described elsewhere [8]. By pulsing one apical nectophore, colonies were capable of rapid turns with a mean angular velocity, ω, of 104 ± 41 deg. s^−1^ and a maximum of 215 ± 90 deg. s^−1^ (Table 1). The maximum measured angular velocity for all colonies was 363 deg. s^−1^. The mean peak acceleration was 9795 ± 7469 deg s^−2^. Turns were characterized by very little forward motion and a tight turning radius: the average length specific radius of curvature, L/R, was 0.15 ± 0.10 (Table 1). As angular velocity increased, R/L also increased (Figure 2), which was indicative of a trade-off between these two performance variables.

### 3.2. Reverse Swimming

Reverse swimming was evoked by stimulating the colony tip, near the pneumatophore, and typically was constrained to a single jet cycle but during more vigorous responses, multiple pulses could be evoked (Figure 3a). During reverse swimming, jet velocities reached a maximum of ~1 m s^−1^, equivalent to 295 nectophore lengths s^−1^, and body speeds reached 134 mm s^−1^ (Table 2), equivalent to 37 nectophore lengths s^−1^. Reynolds number based on nectophore diameter and swimming speed was ~100.

During refill, the velum increased in diameter more quickly than the nectophore and fluid velocities slowed owing to the larger velar opening (Figure 3; Appendix A). Fluid was pulled in posterior to the nectophore and after entering through the velum, the fluid circulated through the nectophore at a relatively low velocity of ~100 mm s^−1^ (Figure 3 and Figure 4), generating vorticity levels of ~250 s^−1^ (Figure 4).

Reversals were accomplished through thrust vectoring of the velum (Figure 5). During forward swimming, the velum was oriented posteriorly at an oblique angle to the colony axis. During reverse swimming, the velum reoriented to direct the jet anteriorly. The typical pattern was for the velum to be oriented either posteriorly (forward swimming) or anteriorly (reverse swimming) for the duration of the jet but in one instance, we observed the velum reorienting mid-cycle (Figure 5, Appendix A). Switching direction mid-pulse is atypical but helps to illustrate the full range of velar orientation.

## 4. Discussion

Moving in three-dimensional space confers ecological advantages to organisms yet is constrained by morphology and hydrodynamics. *N. bijuga* is a colonial invertebrate that is highly proficient at a repertoire of swimming maneuvers. Previous investigations highlighted that coordination of nectophores [7] allows for proficient long-distance migrations [11]. Here we detail that maneuvering and reversal are achieved through rapid changes in the velar aperture and fine tuning of the velar angle to tightly control jet production.

Though cnidarian jellyfish are considered to be primitive forms with simple neurocircuitry [12], individual medusae are highly effective swimmers [13]. Cnidarian medusae represent a diversity of forms with some being adept at rapid swimming and others being more effective at slow but efficient swimming [14,15]. Individual medusae can also effect turns through manipulation of the bell margin or the velum [12,16]. However, individual medusae cannot achieve backwards swimming due to morphological constraints—they have only one jet orifice and the velum has a limited range of motion for thrust vectoring. The arrangement of multiple units in a colony opens up a much richer array of possible maneuvers. Careful quantification of morphology, body motion and structure–fluid interactions are the first steps towards emulating key maneuvering features in an engineering context.

### 4.1. Turning

Jetting kinematics and fluid mechanics during turning are the same as forward swimming but during turning, pulses are constrained to one or two nectophores at the apex of the colony [7]. A long lever arm and resultant torque is achieved by firing the nectophores at the colony apex. In addition to nectophore position, tight orchestration of velar and nectophore movement achieves impressive maneuverability and agility (Table 1). The length-specific radius of *N. bijuga* was 0.15 ± 0.10 with an absolute minimum value of 0.05. Though other organisms such as squid [17] and box fish [18] have R/L values approaching zero, moving forward a tenth or twentieth of a body length during a turn is more than sufficient for maneuvering in most situations. Furthermore, squid and box fish are much more complex than cnidarian swimmers. Aquatic vertebrate swimmers rely on primary and secondary control surfaces along with complex kinematics to move in three-dimensional space [19]. The maximum angular velocity, or agility, of *N. bijuga* was 215 deg. s^−1^ (Table 1), which is higher than that of many vertebrates with more complex body forms and neurocircuitry (Figure 6). Regardless of phylogeny, organisms must produce thrust along a lever arm in order to turn; *N. bijuga* achieves this with a simple linear arrangement of nectophores along a linear axis (Figure 1; Appendix A).

### 4.2. Reversal

During reverse swimming the velum changes orientation to redirect the flow anteriorly (Figure 1 and Figure 5; Appendix A). Remarkably, swimming during reversal matches forward swimming performance as reported in [8]. Jet speed (two-tailed *t*-test, *t* = −1.1; df = 12; *p* = 0.29), body speed (two-tailed *t*-test, *t* = 1.6; df = 11; *p* = 0.14) and jetting time (two-tailed *t*-test, *t* = −1.1; df = 13; *p* = 0.31) are not significantly different between the two swimming modes. Similar performance levels during forward and reverse swimming are likely due to the tight coupling between velum and nectophore kinematics to achieve thrust vectoring. During reverse swimming the velum begins to close in advance of the nectophore (Figure 3) allowing for a narrow, high velocity jet of ~1 m s^−1^ (Table 1) as observed during forward swimming [8]. Having similar levels of performance during forward and reverse motion are rare, even in vertebrates with complex nervous systems. *N. bijuga* achieves matching forward and reverse swimming speeds with a simple nervous system. One way that forward and reverse swimming differ in *N. bijuga* is that refill is slower in reverse swimming (two-tailed *t*-test: *t* = −2.79, df = 13, *p* = 0.015) and as a result the full jet cycle time takes longer during reverse swimming (two-tailed *t*-test: *t* = −2.83, df = 13, *p* = 0.015). Reverse swimming is activated as an escape response and was typically limited to a single pulse (see also [6]). During rare instances of multiple reverse pulses, refill time was comparable to jetting time (Figure 3a). Therefore, refill time is not a factor limiting performance during reverse jetting.

The fluid mechanics of refill, however, are a potential constraint on reverse swimming performance. During refill, the jetting ceases, the colony decelerates and there are potential hydrodynamic losses as fluid is pulled into the nectophore. Solitary medusae have been shown to mitigate these losses during refill through a positive pressure region created by the vortex ring generated underneath the bell [13]. Similarly, *N. bijuga* benefits from a vortex generated inside the nectophore during forward swimming, which generates pressure gradients that enhance forward thrust [8]. During reverse swimming, refill dynamics within the nectophore (Figure 4) match those during forward swimming (Figure 7 in [8]) and would therefore likely counteract reverse thrust. One major difference during refill during forward and reverse swimming is that in forward swimming, fluid is pulled anteriorly to the nectophore and in reverse swimming, fluid is pulled in posteriorly (Figure 4; Appendix A); in each case the fluid is pulled in from a direction that would serve to mitigate losses during refill; i.e., decrease thrust opposing the direction of travel.

Fishes and some planktonic invertebrates are competent at reverse swimming. However, most organisms do not achieve reverse swimming speeds that are comparable in magnitude to their forward swimming speed (Figure 7). *N. bijuga* is notable because the ratio of backward–reverse swimming speed is ~1. This is achieved entirely by simply reorienting the velum anteriorly while the orientation of the rest of the colony and kinematics are unchanged.

### 4.3. Neurological Control

Investigation of kinematics brings up questions about the role of muscle and neuorphysiology in mediating swimming movements. The nectophore is controlled by two sets of muscle fibers—a circular muscle system and a set of radial fibers. The circular muscles control contraction of the nectophore during forward and reverse swimming. The radial fibers of Claus (Claus 1878) contract to pull the velum anteriorly during backward swimming. The fibers of Claus are inactive during forward swimming. Earlier neurophysiological experiments suggested that reverse swimming and forward swimming are controlled by two distinct neurosensory pathways [6]. The forward swimming pathway is excited by stimulation at the base of the colony whereas reverse swimming is excited by stimulation of apical nectophores. The nectophores may experience a sensory transition during development where the younger, apical nectophores evoke reverse swimming and the older nectophores near the colony base evoke forward swimming. Though Mackie [6] suggested that the conduction routes are isolated from one another and that conduction of each is all or none, we made an intriguing observation that the conduction pathway can be interrupted such that the velum reoriented from a posterior directed jet (forward swimming) to an anterior directed jet (reverse swimming) mid-way through a pulse cycle (Figure 5; Appendix A). The coordination of the zooids in the colony requires a more sophisticated set of neural pathways but is still attractive from a bio-emulation perspective. The colony follows a set of simple rules that give rise to a diverse array of maneuvers.

## 5. Conclusions

*N. bijuga* is a highly effective swimmer and a model system for understanding structure–fluid interactions in multi-jet locomotion. It is proficient at long distance swimming but can also perform rapid turns and reversals. Further, effective swimming is achieved with simple cnidarian neurocircuitry. Pelagic colonies are rare, comprising salps and siphonophores, and present unique solutions to aquatic locomotion [27]. The presence of multiple swimming units provides several advantages. Individual units may be small but the colony as a whole can be an order of magnitude larger than any individual unit. Finally, redundancy allows for unaltered whole-colony performance even if individual units are non-functional or lost. The colonial N. *bijuga* is adept at essentially all aspects of swimming and represents an ideal platform for designing a robust underwater vehicle. *N. bijuga* is relatively small with colony lengths of several cm and therefore could inspire microrobots; however, physonect siphonophores occur at a range of sizes—up to 10 m in *Stephanomia* sp. [28]—suggesting the efficacy of multi-jet propulsion at larger scales.

## Figures and Tables

**Figure 1 biomimetics-04-00062-f001:**
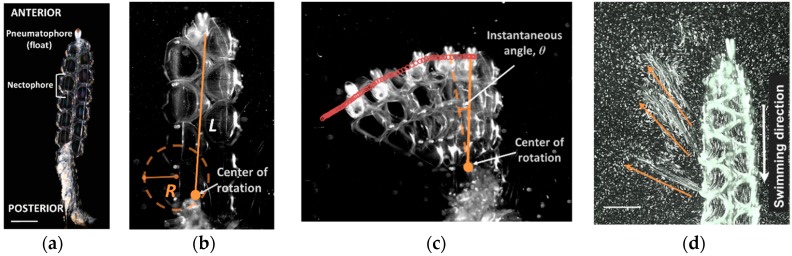
Maneuverability, agility and reverse swimming in the colonial siphonophore, *Nanomia bijuga*. (**a**) Basic anatomy of *N. bijuga*. Scale bar is 3 mm. (**b**) Maneuverability (R/L); R, radius of turning path; L, length of nectosome. (**c**) Agility (angular velocity, ω); *θ*, instantaneous angle. (**d**) Reverse swimming jets. Orange arrows indicate direction of jets. Scale bar is 5 mm. See Appendix A (turning) and Appendix A (reverse).

**Figure 2 biomimetics-04-00062-f002:**
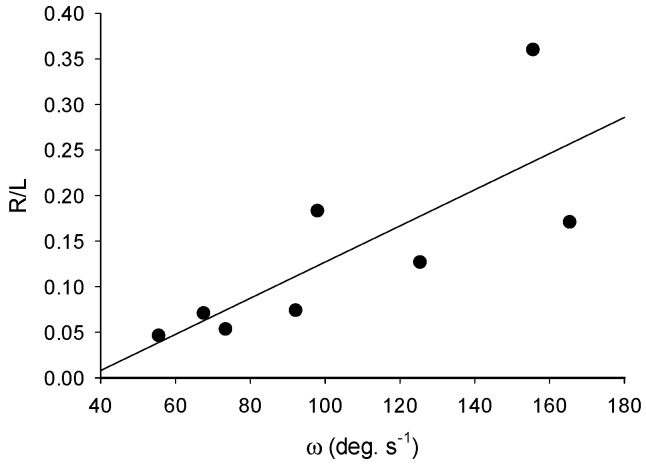
Mean angular velocity, ω, versus length specific radius of turn, R/L; *y* = 0.002*x* − 0.0715; R^2^ = 0.6; *p* = 0.024; *n* = 8.

**Figure 3 biomimetics-04-00062-f003:**
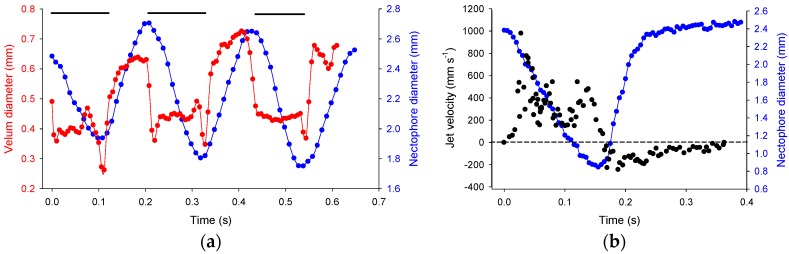
Orchestration of velum and nectophore kinematics and resultant jet velocities during reverse swimming (**a**) Time-varying nectophore diameter and velum diameter over three pulse cycles. Black bars indicate jetting period; (**b**) Time-varying nectophore diameter and jet velocities from particle tracks over one pulse cycle. Jet velocities are negative during refill.

**Figure 4 biomimetics-04-00062-f004:**
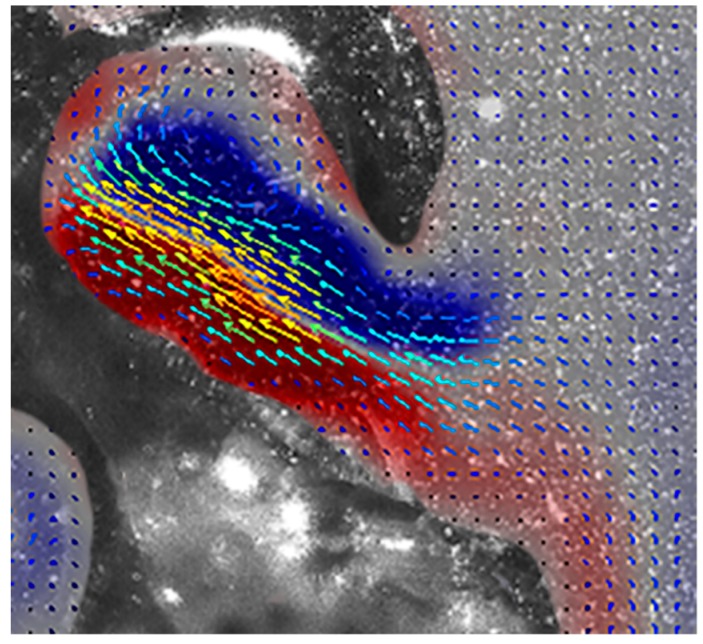
Velocity and vorticity during reverse refill. Fluid is pulled in posterior to the nectophore, potentially enhancing thrust. See Appendix A.

**Figure 5 biomimetics-04-00062-f005:**
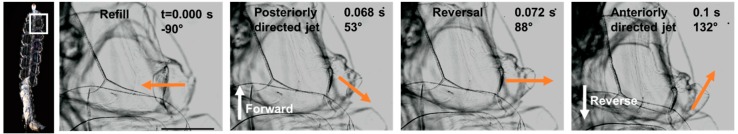
Velum kinematics during transition from forward to reverse swimming with velum angle indicated with orange arrows. Note that a typical jet cycle is either in the forward or reverse direction. Transitions mid-way through a jet are rare but this sequence illustrates the range of velum orientations. White arrows indicate swimming direction. Scale bar in first panel is 1 mm. See Appendix A.

**Figure 6 biomimetics-04-00062-f006:**
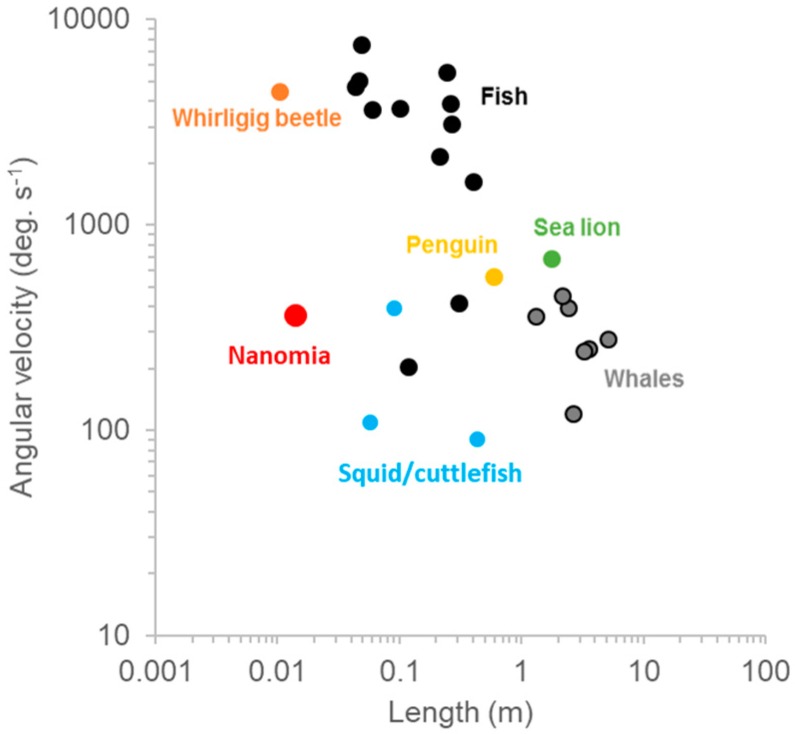
Comparative turning performance in vertebrates and invertebrates. Data are from [17,20] and this study.

**Figure 7 biomimetics-04-00062-f007:**
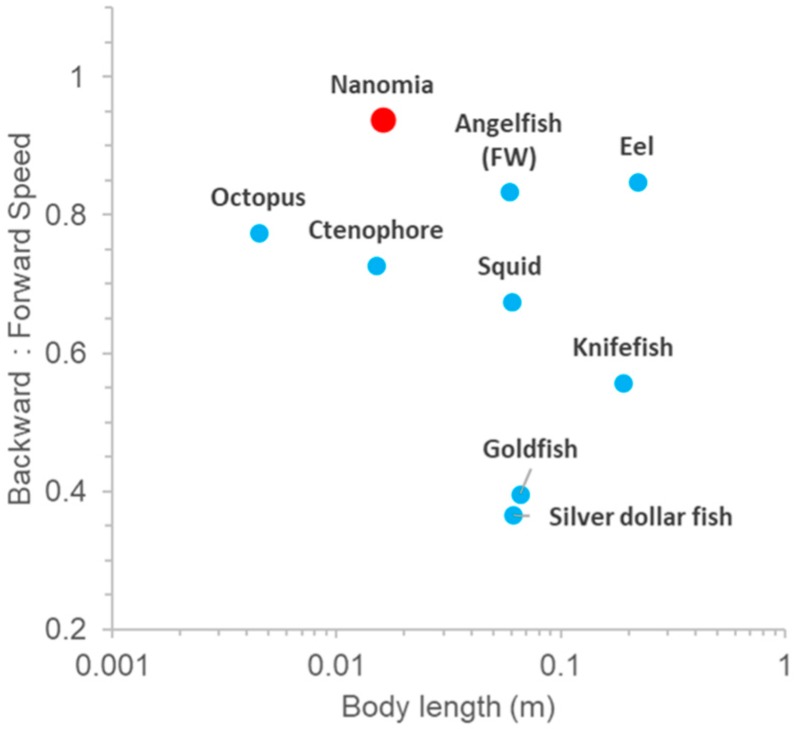
Comparative backward to forward swimming speed ratio in aquatic vertebrates and invertebrates. Squid data show ratio of forward to backward swimming speed since a fins-first, backwards swimming direction is the primary swimming direction. Data are from [21,22,23,24,25,26] and this study.

**Table 1 biomimetics-04-00062-t001:** *N. bijuga* turning performance. ω, angular velocity; Acc, acceleration; *R*, radius of curvature; *R*/L, length specific turning radius. Mean values ± st. dev.; max. and min. values are based on the highest and lowest mean values from all sequences; N, number of colonies; # seq., number of sequences.

	ω (deg s^−1^)	ω_max_ (deg s^−1^)	Acc._max_ (deg s^−2^)	R (mm)	R/L
Mean	104 ± 41	215 ± 90	9795 ± 7469	1.64 ± 0.77	0.15 ± 0.10
Max	165	363	21635	3.10	0.36
Min	56	122	3510	0.72	0.05
N	6	6	6	7	9
# seq.	8	8	8	11	11

**Table 2 biomimetics-04-00062-t002:** *N. bijuga* reverse swimming performance. N, number of colonies; # seq., number of sequences.

	Jetting (s)	Refilling (s)	Total (s)	Ratio	Max. Jet Speed (mm s^−1^)	Max. Body Speed (mm s^−1^)
Mean	0.14 ± 0.02	0.23 ± 0.08	0.37 ± 0.10	0.67	1063 ± 176	134 ± 28
Max	0.10	0.10	0.21	0.43	879	98
Min	0.16	0.30	0.44	0.99	1260	166
N	6	5	5	5	5	4
# seq.	6	5	5	5	5	5

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
