# Peer review of "Maneuvering Performance in the Colonial Siphonophore, *Nanomia bijuga"

_biomimetics, 2019, doi:10.3390/biomimetics4030062_

Round 1
Reviewer 1 Report
It's crucial to have a picture and diagram of Nanomia identifying relevant morphology and the various directions of jetting described in the paper. The longitudinal axis of the animal should be identified, and anterior and posterior identified. In particular the label "reverse swimming" should be reviewed and changed with relation to the morphology of the animal. Line 230 appears to refer to a negative velocity. I suppose this is intended to refer to so-called "reverse swimming" but only serves to highlight the confusion that can arise when proper morphological axes are not observed.
All the figures must be referred back to this master diagram, showing anatomy and orientation of the morphological components.
Are nectophores specialised to work in limited directions, or can any nectophore contribute to thrust for any maneuver or direction of swimming? Please label if so.
The last sentence of the paper is, presumably, the tag for publishing in “Biomimetics”.This sentence should either be expanded into a paragraph to outline scaling factors involved in this form of locomotion and maneuvering or omitted altogether.
Author Response
Point 1:
It's crucial to have a picture and diagram of Nanomia identifying relevant morphology and the various directions of jetting described in the paper. The longitudinal axis of the animal should be identified, and anterior and posterior identified. In particular the label "reverse swimming" should be reviewed and changed with relation to the morphology of the animal. Line 230 appears to refer to a negative velocity. I suppose this is intended to refer to so-called "reverse swimming" but only serves to highlight the confusion that can arise when proper morphological axes are not observed.
All the figures must be referred back to this master diagram, showing anatomy and orientation of the morphological components.
Response 1:
We have added a panel in Fig. 1 to clearly label the basic anatomy and anterior/posterior orientation. We have also added an arrow in Fig. 1 D to indicate swimming direction. We edited the text throughout to use more consistent terminology. During forward swimming, the velum and jet are oriented posteriorly and during reverse swimming, the velum and jet are oriented anteriorly. We updated Fig. 5 to reflect this terminology. We agree with the reviewer that this will help clarify the different swimming orientations. Point 2:
Are nectophores specialised to work in limited directions, or can any nectophore contribute to thrust for any maneuver or direction of swimming? Please label if so.
Response 2:
Any nectophore can contribute to thrust during forward swimming though the relative contributions depend on the location in the colony as we’ve described in the text (Lines 51-67); also detailed in Costello et al 2015). Any nectophore can contribute to thrust during reverse swimming by reorienting the velum anteriorly. Turning is achieved by firing the anterior-most nectophores. We modified the text to make these distinctions more clear (Lines 67-69).
Point 3:
The last sentence of the paper is, presumably, the tag for publishing in “Biomimetics”.This sentence should either be expanded into a paragraph to outline scaling factors involved in this form of locomotion and maneuvering or omitted altogether.
Response 3:
Our view is that we have referred to the unique features of N. bijuga that make it a desirable target in an engineering context throughout the text (please see, for example, Lines 41-43; 81-85; 251-253; 331-334). We have also added a paragraph to the Introduction discussing current maneuvering limitations of underwater vehicles (Lines 46-53) and some text at the end of the Conclusion discussing the scalability of multi-jet systems.
Reviewer 2 Report
The manuscript deals with an analysis about the kinematics and fluid mechanics of Nanomia Bijuga. This topic is interesting for the scientific community and the text is well-written. For this reason, I recommend to accept the manuscript after realizing the following minor modifications:
- The introduction must be extended providing more related studies and indicating grey aspects that must be studied in greater detail.
- In Section 2.1 it is necessary to include more information about the characteristics of the equipment employed.
- There is a problem with the text of the x axis of Fig. 2
Best regards
Author Response
Response to Reviewer 2 comments
The manuscript deals with an analysis about the kinematics and fluid mechanics of Nanomia Bijuga. This topic is interesting for the scientific community and the text is well-written. For this reason, I recommend to accept the manuscript after realizing the following minor modifications:
Point 1:
- The introduction must be extended providing more related studies and indicating grey aspects that must be studied in greater detail.
Response 2:
We have added a paragraph to the Introduction discussing the advantages of pulsed jets and the potential for multi-jet systems to overcome current maneuvering limitations of underwater vehicles (Lines 46-53).
Point 2:
- In Section 2.1 it is necessary to include more information about the characteristics of the equipment employed.
Response 2:
We have added more information about the equipment. The data were collected during three different visits and we used multiple models of cameras (now indicated) and lasers but the optical set-up—which we describe in the text— was always the same.
Point 3:
- There is a problem with the text of the x axis of Fig. 2
Response 3:
The Fig. 2 x-axis text looks fine in the version we received from the editor. Perhaps this was an issue with a special character that didn’t come through properly in the reviewer’s version of Word? We’d be happy to make a change if needed.